# New Insights into Adipokines in Gestational Diabetes Mellitus

**DOI:** 10.3390/ijms23116279

**Published:** 2022-06-03

**Authors:** Jorge Valencia-Ortega, Rebeca González-Reynoso, Edgar G. Ramos-Martínez, Aldo Ferreira-Hermosillo, María I. Peña-Cano, Enrique Morales-Ávila, Renata Saucedo

**Affiliations:** 1Unidad de Investigación Médica en Enfermedades Endocrinas, UMAE Hospital de Especialidades, Centro Médico Nacional Siglo XXI, Instituto Mexicano del Seguro Social, Mexico City 06720, Mexico; j.valencia.o@hotmail.com (J.V.-O.); alexaitzelrebeca@gmail.com (R.G.-R.); aldo.nagisa@gmail.com (A.F.-H.); 2Facultad de Química, Universidad Nacional Autónoma de México, Mexico City 04510, Mexico; edgargus2@gmail.com; 3Hospital de Ginecología y Obstetricia 221, Instituto Mexicano del Seguro Social, Toluca 50090, Mexico; isabelpenacano@hotmail.com; 4Facultad de Química, Universidad Autónoma del Estado de México, Toluca 50120, Mexico; enrimorafm@yahoo.com.mx

**Keywords:** gestational diabetes, adipokines, adipose tissue, placenta, umbilical cord, insulin resistance, inflammation

## Abstract

Gestational diabetes mellitus (GDM) is the most common metabolic disorder of pregnancy and has considerable short- and long-term consequences for the health of both the mother and the newborn. Within its pathophysiology, genetic, nutritional, epigenetic, immunological, and hormonal components have been described. Within the last two items, it is known that different hormones and cytokines secreted by adipose tissue, known collectively as adipokines, are involved in the metabolic alterations underlying GDM. Although the maternal circulating profile of adipokines in GDM has been extensively studied, and there are excellent reviews on the subject, it is in recent years that more progress has been made in the study of their expression in visceral adipose tissue (VAT), subcutaneous adipose tissue (SAT), placenta, and their concentrations in the umbilical circulation. Thus, this review compiles and organizes the most recent findings on the maternal and umbilical circulating profile and the levels of expression of adipokines in VAT, SAT, and placenta in GDM.

## 1. Introduction

Gestational diabetes mellitus (GDM) is the most common metabolic disorder of pregnancy. It is defined as any degree of glucose intolerance with onset in the second or third trimester of pregnancy, and which is clearly not pre-existing diabetes [1]. Due to ethnic characteristics and the use of different diagnostic criteria, the prevalence of this disorder varies globally, with figures ranging from 1% to over 30% [2]. Worldwide, these figures have increased, along with the increase in the prevalence of obesity in women in the reproductive stage. Other risk factors for GDM are advanced maternal age, parity, family history of type 2 diabetes mellitus (DM2), personal history of GDM or macrosomia in previous pregnancies, and excessive weight gain in pregnancy. Although glucose intolerance usually resolves after delivery, women with GDM have a seven-fold increased risk of developing DM2 in the future compared to women with euglycemic pregnancy. The risk of atherosclerosis is also significantly increased in these women [3,4]. In addition, the neonates of women with GDM have an increased risk of acute perinatal complications such as hypoglycemia, jaundice, large for gestational age and shoulder dystocia. Moreover, the risk of developing obesity, hypertension, diabetes, and cardiovascular disease in adulthood is increased [5,6,7].

Although some key pathophysiological mechanisms in GDM have been extensively described to date, it is in the last few decades that several adipokines have shown some involvement in the metabolic alterations underlying GDM. Today, there are excellent reviews of the role of adipokines in GDM, with special focus on the maternal circulating profile [8,9,10,11]. However, it is in recent years that more progress has been made in the study of their expression in visceral adipose tissue (VAT), subcutaneous adipose tissue (SAT), placenta, and their concentrations in the umbilical circulation. Thus, this review compiles and organizes the most recent findings on the maternal and umbilical circulating profile and the levels of expression of adipokines in VAT, SAT, and placenta in GDM. Figure 1 summarizes the main findings on adipokine dysregulation in GDM.

## 2. Methods

Relevant English language articles were searched in Pubmed and Embase using the keywords “gestational diabetes”, “adipokines”, “adipose tissue”, “placenta”, “umbilical cord”, “insulin resistance”, and “inflammation”. It is important to highlight that with the aim of compiling the most recent studies, only those published from 2017 to 2022 that evaluated adipokines in maternal circulation were included. For the section on adipokines in adipose tissue, placenta, and umbilical circulation, we included papers published between 2010 and 2022 (since there are few studies on the subject).

## 3. Adipokines in Maternal Circulation

Table 1 shows the main characteristics of the most recent articles on maternal concentrations of adiponectin, leptin, resistin, tumor necrosis factor alpha (TNF-α), interleukin 6 (IL-6), interleukin 8 (IL-8), interleukin 10 (IL-10), interferon-gamma (IFN-γ), adipsin, neutrophil gelatinase associated lipocalin (NGAL), nerve growth factor (NGF), monocyte chemoattractant protein-1 (MCP-1), omentin-1, fatty acid binding protein-4 (FABP4), chemerin, soluble leptin receptor (sOB-R), vaspin, retinol binding protein-4 (RBP-4), visfatin, plasminogen activator inhibitor-1 (PAI-1), angiopoietin-like protein 8 (ANGPTL8), nesfatin-1, delta-like 1 (DLK1), fetuin A, fetuin B, and adipocyte fatty acid binding protein (AFABP) in GDM.

In summary, of the eight cohort studies that analyzed circulating levels of adiponectin, seven agree that its levels are decreased in the first and second trimester, including also in the early third trimester, but not in the late third trimester, of women with GDM, and only one did not observe differences in the first and second trimester. The four case-control studies reviewed agree with these findings, as three found lower adiponectin levels in the second trimester, and one observed no difference in late pregnancy. Regarding leptin, of the six cohort studies, three reported that concentrations are higher in women with GDM in the first and second trimester, including also in the early third trimester, but not in the late third trimester, and three did not observe differences in the first and second trimester. Except for Zhang, et al., the case-control study reviewed for leptin did not observe differences in its concentrations at the end of pregnancy between GDM and controls. Three cohort studies and one case-control study indicated that resistin levels were not different between GDM and controls throughout pregnancy. A cohort study for TNF-α did not observe differences in the first and second trimester, but another cohort study showed that concentrations of this cytokine are higher in the third trimester in GDM compared with euglycemic pregnancy, and of the three case-control studies that measured its concentrations at the end of pregnancy, two reported higher levels in GDM than in controls, while the other did not observe differences. The three cohort studies that measured IL-6 did not observe a difference in the first, second and third trimester, nor did the case-control study that measured the concentrations of this cytokine at the end of pregnancy. In relation to NGAL, a cohort study reported elevated levels in the first trimester in women with GDM, and a case-control study observed that levels were higher in GDM at the end of pregnancy, but another study with the same design observed that concentrations were similar between groups. Two cohort and two case-control studies indicated that omentin-1 levels were not different between GDM and normal glucose tolerance (NGT) throughout pregnancy. The four studies on ANGPTL8 suggest that its concentrations are higher in women with GDM in all trimesters of gestation. Three studies showed that nesfatin-1 levels at gestational weeks 24–28 are lower in women with GDM than in controls. Regarding IL-8, IL-10, IFN-γ, adipsin, NGF, MCP-1, FABP4, chemerin, sOB-R, vaspin, RBP-4, visfatin, PAI-1, DLK1, fetuin A, fetuin B, and AFABP studies are few or differ in their findings, so it is not possible to highlight any trend.

There are some studies with a peculiar design, for example, the cross-sectional study by Jeon et al., with 57 women with GDM classified according to pre-pregnancy body mass index (BMI), maternal age, and gestational weight gain at the time of GDM diagnosis. Among the objectives of the study was to compare circulating levels of leptin, resistin, and adiponectin at 24–28 weeks of gestation. It was observed that leptin concentrations were significantly higher in women with GDM who had a pre-pregnancy BMI ≥ 25 kg/m^2^ than in those with a BMI < 25 kg/m^2^. A significant correlation was identified between leptin concentrations and BMI pre-pregnancy and at the time of GDM diagnosis. No differences or correlations were observed in resistin and adiponectin measurements [37]. Another example is the study by Tsiotra et al., in which circulating levels of visfatin, omentin-1, chemerin, RBP4, resistin, adiponectin, leptin, TNF-α, and IL-6 were measured in 23 women with NGT and 15 with GDM, all with a resolution of pregnancy at term, and classified as obese (BMI ≥ 30 kg/m^2^) and non-obese (BMI < 30 kg/m^2^). Obese GDM women showed higher chemerin and lower visfatin levels compared to non-obese NGT women. Obese and non-obese GDM women showed lower levels of omentin-1 compared to the non-obese NGT group. Leptin levels were significantly elevated in obese GDM and obese NGT compared with non-obese GDM and non-obese NGT. The adiponectin/leptin ratio was significantly lower in obese GDM and obese NGT compared with non-obese GDM and non-obese NGT. TNF-α levels were higher in all obese compared with non-obese women. Both pre-pregnancy and gestational BMI correlated positively with leptin and TNF-α levels, and negatively with omentin-1 and adiponectin/leptin ratio. In addition, chemerin levels correlated positively with gestational BMI, Homeostatic Model Assessment-Insulin Resistance (HOMA-IR), and glucose, and leptin levels with HOMA-IR, and insulin. The adiponectin/leptin ratio correlated negatively with insulin concentrations [38].

Condensing the evidence on adipokines in maternal circulation, we can state that GDM is characterized by elevated concentrations of leptin, TNF-α, NGAL and ANGPTL8, and decreased concentrations of adiponectin and nesfatin-1, although it should be noted that gestational age may have an important influence on these concentrations. Based on the idea that these circulating adipokines are able to exert their effects on insulin-sensitive tissues, the following describes how they influence insulin sensitivity.

Leptin is a 16 kDa protein hormone secreted mainly by adipocytes, which participates in the central regulation of food intake and energy expenditure to maintain body fat stores. It exerts this effect mainly through the leptin receptor OB-Rb in the hypothalamic arcuate nucleus, but there are five other isoforms of this receptor (OB-Ra, OB-Rc, OB-Rd, OB-Re, and OB-Rf) distributed in different tissues [39,40]. Physiologically, leptin levels increase as pregnancy progresses, and are thought to be responsible for the changes in energy balance associated with pregnancy [41]. There is strong evidence that the placenta, rather than maternal adipose tissue, contributes to the increase of maternal leptin concentrations during pregnancy [42]. This adipokine directly affects whole body insulin sensitivity by regulating the secretion of insulin, and by hepatic regulation of gluconeogenesis [43]. The mechanism of leptin contribution to the development of insulin resistance may lie in its capacity to phosphorylate insulin receptor 1 substrate (IRS1) serine residues, which downregulates insulin signaling. Moreover, it exerts an acute inhibitory effect on insulin secretion by pancreatic β-cells [44,45].

TNF-α is a 17 kD homotrimeric cytokine that has pleiotropic effects on various cell types. It is mainly secreted by immune cells such as activated macrophages, T cells, and natural killer cells; however, many different types of cells in the body can produce it. In general, TNF-α binds to its receptors, mainly TNFR1 and TNFR2, and then transmits molecular signals for biological functions such as inflammation and cell death. TNFR1 is expressed by all human tissues and is the key signaling receptor for TNF-α, while TNFR2 is generally expressed in immune cells and facilitates limited biological responses [46]. TNF-α plays a critical role in developing insulin resistance by reducing the insulin-regulated glucose transporter type 4 (GLUT4) expression, located in adipocytes, skeletal, and cardiac muscles, and through the induction of serine phosphorylation of IRS1 [47].

NGAL is a novel adipokine highly expressed in adipose tissue, and its increased concentrations are associated with obesity, insulin resistance and diabetic nephropathy [48]. Although two cross-sectional studies reported increased levels of NGAL in GDM in late pregnancy, it is not known how this adipokine is related to glucose metabolism and insulin sensitivity. NGAL has been shown to induce apoptosis in beta cells and it has been proposed that it may induce insulin resistance indirectly by promoting inflammation [49].

ANGPTL8, a 22 kDa protein, is secreted by the liver and adipose tissue and plays important roles in glucose homeostasis, lipid metabolism and inflammation [50]. Today there is a discussion on whether ANFPTL8 is able to induce pancreatic beta-cell proliferation and increase insulin secretion, although the consensus is that it is not capable of such actions [10]. Many studies have shown that ANGPTL8 correlates negatively with HDL-C levels and positively with triglyceride levels, making it a crucial modulator of lipid metabolism [51]. The role of ANGPTL8 in glucose and lipid homeostasis calls for more investigation.

Adiponectin is a protein of 244 amino acids with a molecular weight of 32 kDa secreted mainly by white adipose tissue, although other tissues also produce it, but at low levels [52]. It is synthesized by the adipocyte into circulation in four forms: trimers, hexamers, high molecular weight multimers and the globular form, which results from proteolytic cleavage at the amino acid 110 [53,54]. Adiponectin has three receptors: (1) AdipoR1, a high-affinity receptor for globular adiponectin with low affinity for the full-length form, with ubiquitous expression, although more abundant in skeletal muscle; (2) AdipoR2, which mainly recognizes the full-length form and is predominantly expressed in the liver; and (3) T-cadherin, which acts as a receptor for the hexameric and multimeric forms [55,56]. Most studies indicate that circulating levels of adiponectin decrease as pregnancy progresses [57]. Adiponectin has effects at the adipose tissue level, where it participates in adipocyte differentiation, suppression of inflammation, and the lipotoxic effects of lipid accumulation. In the liver, it suppresses gluconeogenesis, which lowers serum glucose levels, and in the skeletal muscle it mainly exerts its beneficial metabolic effects [52]. The latter is supported by several studies, with mouse and rat myocytes showing that adiponectin promotes glucose transporter 4 (GLUT4) translocation, glucose utilization and fatty acid oxidation [58,59,60]. In humans, adiponectin has been suggested to have anti-inflammatory, anti-diabetic, and anti-atherogenic functions, which makes sense with low circulating levels of this adipokine in conditions of obesity, insulin resistance, and inflammation [61,62,63].

Nesfatin-1 is a protein of 82 amino acids secreted by the hypothalamus and peripheral tissues such as the adipose tissue, pancreas and duodenum. One of the key functions of nesfatin-1 is the regulation of glucose metabolism. Nesfatin-1 may enhance insulin sensitivity and increase glucose uptake in peripheral tissues via Akt/AMPK/TORC2, and may suppress hepatic gluconeogenesis through inhibition of the mTOR-STAT3 pathway [64,65]. In addition, higher nesfatin-1 levels enhances glucose-induced insulin secretion by stimulating Ca^2+^ influx through L type channels [66].

Figure 2 summarizes the mechanisms of these circulating adipokines in GDM.

## 4. Adipokines in Visceral and Subcutaneous Adipose Tissue

Until three decades ago, it was thought that white adipose tissue only functioned as an energy store, thermal insulator, and mechanical protector; but today it is known to have an important endocrine function through the secretion of soluble mediators known collectively as adipokines. These are a group of protein hormones and cytokines secreted by adipocytes, immune system cells, fibroblasts, and vascular cells that regulate the local and systemic activity of the organism through their participation in glucose and lipid metabolism, insulin sensitivity, appetite, immune response, and inflammation [8,67,68].

Dysregulation of adipokine production is a characteristic of obesity and obesity-related diseases [69]. This is explained by the fact that excessive body weight promotes adipocyte hypertrophy and local hypoxia. This causes the development of a chronic inflammatory state that recruits macrophages and lymphocytes that, in turn, alter the balance between pro- and anti-inflammatory cytokines, thus originating a stressor environment that affects the adipokine synthesis profile [70,71]. It has been observed that this adipokine profile is strictly related to excess body weight, percentage of hormonally active adipose tissue, and inflammatory status, and that these changes can be reversed by reducing VAT and SAT levels [72,73,74].

In relation to excessive weight in pregnancy, women who are overweight or obese are at greater risk of presenting obstetric complications, such as hypertensive disorders and GDM [75,76]. Alarmingly, in Mexico, nearly 75% of women of reproductive age (20–49 years old) are obese or overweight, while in the United States they represent more than 60% [77]. There are several hypotheses on the relationship between overweight and obesity with the development of obstetric complications, among which the pathophysiological contribution of adipokines stands out, which seems promising in the identification of early biomarkers and new therapeutic targets [78].

Table 2 shows the main characteristics of the articles on gene and/or protein expression in adipose tissue of adiponectin, leptin, omentin-1, resistin, IL-1β, IL-6, IL-1RA, IL-10, TNF-α, SOCS3, visfatin, apelin, adrenomedullin (ADM), and nesfatin-1 in GDM.

In the study by Tsiotra et al., mentioned above, gene expression levels of visfatin, omentin-1, chemerin, and leptin were also analyzed in VAT and SAT. It was observed that, in VAT, obese women with GDM had higher levels of chemerin mRNA compared with non-obese controls. Leptin expression in VAT of obese controls and obese GDM women was higher than in non-obese controls. Considering all women, leptin mRNA levels in VAT correlated with glucose concentrations and gestational BMI, whereas its levels in TAS correlated positively with gestational BMI [38]. Another study with a peculiar design is the investigation by Li et al., who analyzed chemerin mRNA levels in SAT of 10 pregnant women with GDM and 10 controls classified according to their weight, and observed that chemerin expression levels were lower in obese controls, normal-weight women with GDM, and obese women with GDM, compared to normal-weight controls [86].

It is important to highlight two aspects related to the study of adipokines in adipose tissue: (1) more research is needed in the study of SAT; (2) in most studies, mRNA or protein levels of adipokines did not correlate with clinical or biochemical variables, so it is better to think that these mediators mainly have a local effect. Taking these considerations into account, we can declare that GDM is characterized by elevated expression of leptin, resistin, TNF-α, SOCS3, ADM and decreased expression of adiponectin in VAT.

Studies suggest that, in the adipocyte, leptin inhibits insulin-stimulated glucose uptake and that high concentrations of leptin induce the expression of SOCS3 which inhibits insulin receptor autophosphorylation and tyrosin phosphorylation of IRS1 [87,88,89]. In this same cell, TNF-α decreases the expression of insulin receptor, IRS1 and GLUT4, and stimulates the production of SOCS3 [89,90,91,92]. Regarding resistin, its overexpression in 3T3-L1 adipocytes increases the levels of TNF-α and inhibits GLUT4 activity and its gene expression, reducing insulin’s ability for glucose uptake [93]. Interestingly, in VAT explants from normal-weight women, TNF-α increases gene expression of ADM, CRLR, RAMP2, and RAMP3 in a dose-dependent manner, and in human adipocyte cell line, ADM stimulates glycerol release into the medium in a dose-dependent manner, which is reversed by ADM antagonist, indicating that ADM directly stimulates lipolysis [79]. On the other hand, adiponectin acts locally on the adipocyte to increase glucose uptake through increased phosphorylation and catalytic activity of AMP-activated protein kinase (AMPK) [94].

Figure 3 summarizes the mechanisms of these adipokines in VAT of GDM women.

## 5. Adipokines in Placenta and Cord Blood

Historically, the placenta is thought to play an important role in the physiological insulin resistance of pregnancy, through the production of placental hormones such as human placental lactogen and placental growth hormone, as their effects can interfere with insulin receptor signaling and cause a marked decrease in glucose utilization. It is known that GDM affects placental functions and structures from the early stages of the disease. It has been observed that in this disorder, the placenta shows increased weight, glycogen deposits, increased number of syncytial knots, villous edema, increased angiogenesis and vasculogenesis, increased fibrinoid necrosis, chorangiosis, ischemia, altered lipid metabolism, and exacerbated mitochondrial dysfunction [95]. Under these conditions, the placenta responds by synthesizing various cytokines, chemokines, hormones, and adipokines that can migrate into the maternal and umbilical circulation and are capable of contributing to the pathophysiological mechanisms of the disorder [96,97,98,99].

Table 3 shows the main characteristics of the articles on mRNA and/or protein levels in placenta and umbilical cord of TNF-α, leptin, apelin, NGAL, adiponectin, resistin, visfatin, omentin-1, fetuin A, RBP4, nesfatin-1, AFABP, vaspin, and irisin in GDM.

In the study by Tsiotra et al., they also analyzed gene expression levels of visfatin, omentin-1, chemerin, and leptin in the placenta, and found that visfatin mRNA levels were three times lower in obese GDM than in non-obese GDM women. Furthermore, considering all women, levels of placental chemerin mRNA were negatively correlated with the weight of the newborn [38].

Condensing the evidence on adipokines in placenta and umbilical cord, we can state that GDM is characterized by elevated placental expression of TNF-α, NGAL, resistin, and visfatin, and increased levels of leptin, TNF-α, NGAL, RBP4, visfatin, and decreased concentrations of adiponectin, AFABP, and fetuin A in umbilical cord.

It is evident that further research on placental and umbilical cord adipokines is needed, focusing mainly on their placental and neonatal effects, since studies do not show relevant correlations of placental and umbilical cord adipokine levels with maternal clinical and biochemical features in GDM. It should be noted that although six different types of GLUT are expressed in the placenta, including GLUT4, insulin-independent GLUT1 is the most abundant isoform, and is therefore considered to be the main responsible for the maternal-fetal glucose exchange [109], so it is likely that the placental effects of these adipokines are not directly related to insulin resistance. In line with this, it has been observed that macrosomia is not associated with changes in the placental expression of GLUT isoforms [110,111]. Interestingly, umbilical cord concentrations of leptin, adiponectin, and resistin correlate with newborn birth weight [100], and the leptin/adiponectin ratio correlates positively with insulin resistance levels and negatively with insulin sensitivity levels [101], suggesting that these adipokines regulate newborn glucose metabolism.

## 6. Conclusions

Several studies have investigated changes in adipokine levels in the maternal circulation of women with GDM. In comparison, few studies have evaluated adipokine levels in umbilical circulation, SAT, VAT, placenta and cord blood. Most authors have observed that, from the first trimester to the early third trimester, adiponectin concentrations are decreased and leptin concentrations are increased in women with GDM, and correlate with data of insulin resistance, altered lipid metabolism, and excess body weight. Notably, both adiponectin and leptin concentrations from the first to the second trimester are significant predictors of GDM risk. On the other hand, more studies longitudinally assessing the maternal levels of IL-8, IL-10, IFN-γ, adipsin, NGAL, NGF, MCP-1, omentin-1, FABP4, chemerin, sOB-R, vaspin, RBP-4, visfatin, PAI-1, ANGPTL8, nesfatin-1, DLK1, fetuin A, fetuin B, and AFABP are needed to have a better understanding of their pathophysiological roles in GDM. Regarding the umbilical concentrations of adipokines and their levels in TAS, VAT, placenta, and umbilical cord, the cross-sectional assessment at the end of pregnancy that all studies have, and which is justified by the inaccessibility of these sites during pregnancy, does not allow us to identify their pathophysiological relevance, since most probably these levels vary throughout pregnancy in GDM. Finally, the authors consider that it would be very valuable to longitudinally evaluate the maternal levels of both adipokines whose pathophysiological relevance is well established, and those of the less-studied adipokines, in order to have a more solid perspective of their pathophysiological relevance in GDM.

## Figures and Tables

**Figure 1 ijms-23-06279-f001:**
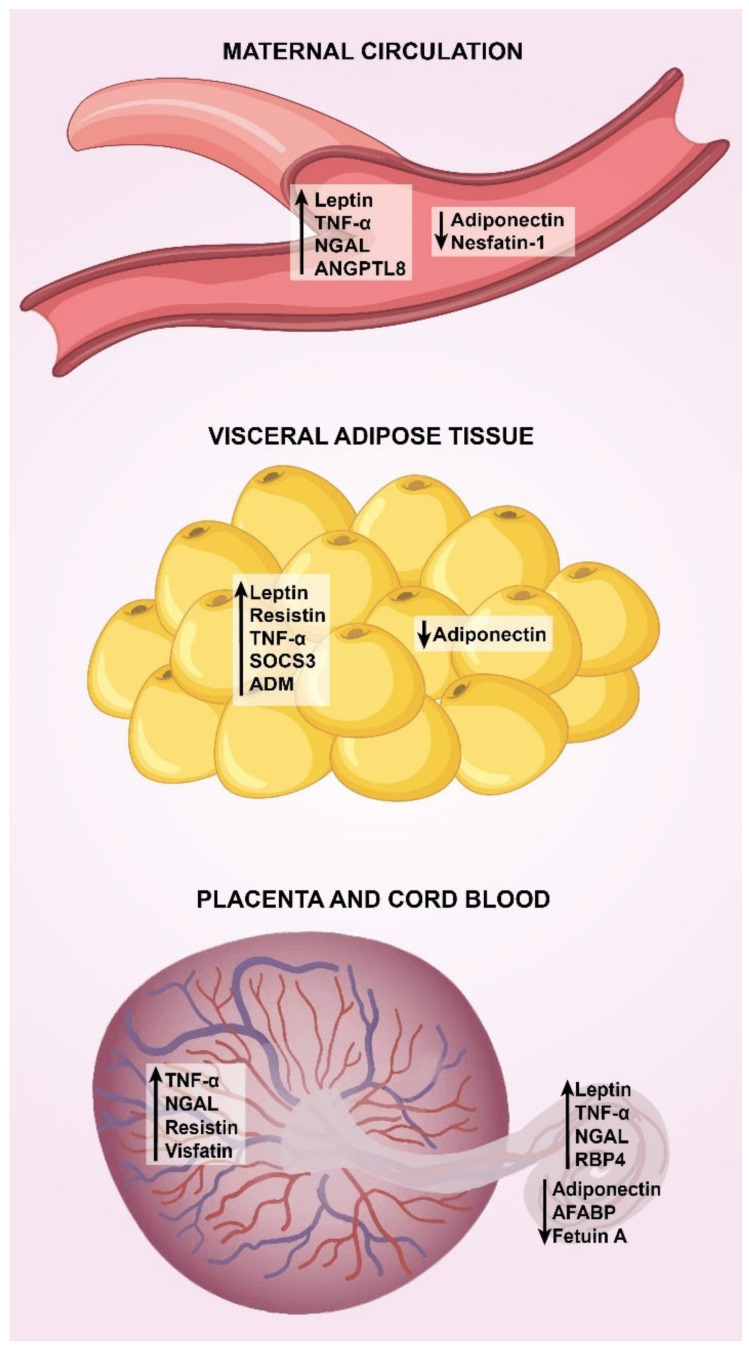
**Main adipokines deregulated in GDM.** Major trends in adipokine dysregulation in GDM are shown in maternal circulation, visceral adipose tissue, placenta and umbilical circulation (see the text for deeper details). In maternal circulation and cord blood, arrows indicate increased or decreased concentrations. In visceral adipose tissue and placenta, arrows indicate up-regulation or down-regulation. TNF-α, tumor necrosis factor alpha; NGAL, neutrophil gelatinase associated lipocalin; ANGPTL8, angiopoietin-like protein 8; SOCS3, suppressor of cytokine signaling 3; ADM, adrenomedullin; RBP-4, retinol binding protein-4; AFABP, adipocyte fatty acid-binding protein.

**Figure 2 ijms-23-06279-f002:**
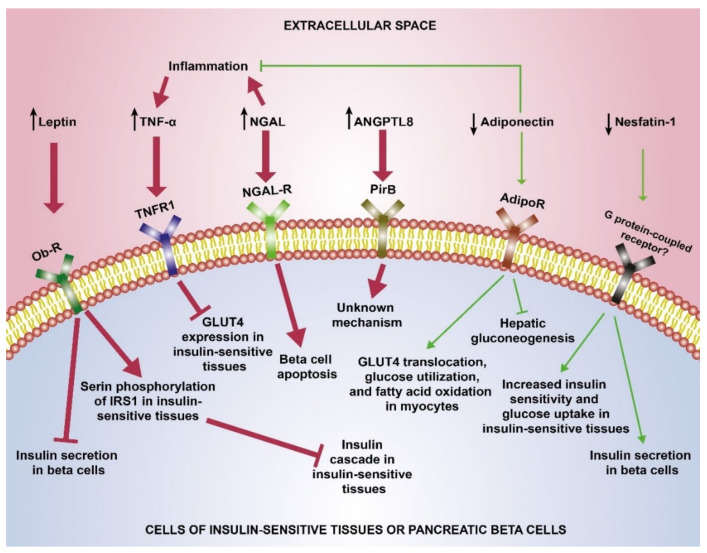
**Effects of deregulated maternal circulating adipokines in GDM.** Potential effects of leptin, TNF-α, NGAL, ANGPTL8, adiponectin, and nesfatin-1, and their respective receptors, in insulin-sensitive tissues or pancreatic beta cells in GDM. Elevated concentrations of leptin, TNF-α, NGAL, and ANGPTL8 potentiate red pathways that promote a hyperglycemic state, whereas decreased concentrations of adiponectin and nesfatin-1 weaken green pathways that favor a euglycemic state (see the text for deeper details). Black arrows indicate increased or decreased concentrations. In the colored arrows, the arrowhead lines indicate stimulation and the T-shaped lines indicate inhibition. TNF-α, tumor necrosis factor alpha; NGAL, neutrophil gelatinase associated lipocalin; ANGPTL8, angiopoietin-like protein 8; Ob-R, leptin receptor; TNFR1, tumor necrosis factor receptor 1; NGAL-R, neutrophil gelatinase associated lipocalin receptor; PirB, paired immunoglobulin-like receptor B; AdipoR, adiponectin receptor; IRS1, insulin receptor 1 substrate; GLUT4, glucose transporter type 4.

**Figure 3 ijms-23-06279-f003:**
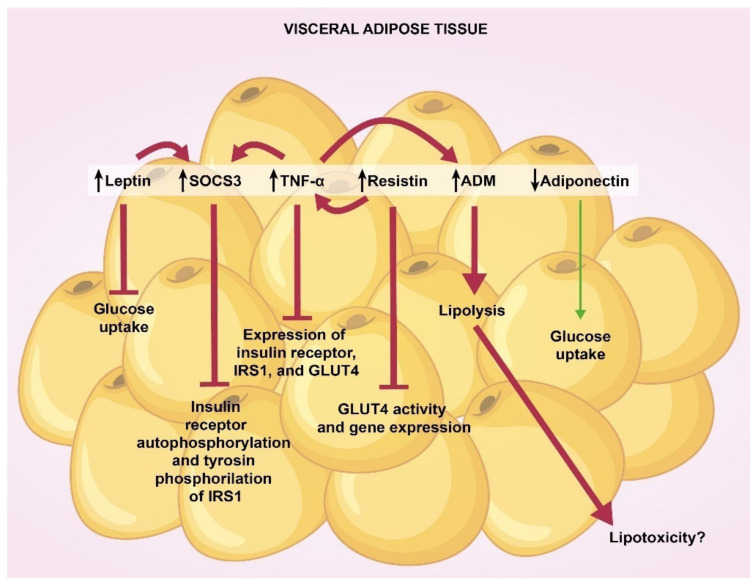
**Effects of visceral fat adipokines GDM.** Elevated expression of leptin, SOCS3, TNF-α, resistin, and ADM potentiate red pathways that promote insulin resistance, whereas decreased concentrations of adiponectin weaken green pathways that favor glucose uptake (see the text for deeper details). SOCS3, suppressor of cytokine signaling 3; TNF-α, tumor necrosis factor alpha; ADM, adrenomedullin; IRS1, insulin receptor 1 substrate; GLUT4, glucose transporter type 4.

**Table 1 ijms-23-06279-t001:** Characteristics and findings of studies comparing maternal circulating concentrations of adipokines between GDM patients and controls.

Authors (Year)	Study Design	*n*	Gestational Age at Measurement	Adipokine Concentration in GDM	Secondary Findings and Comments
Adiponectin					
Tagoma, et al. (2022) [12]	Cohort	213 (60 GDM and 153 controls)	Weeks 23–28	↓	Not reported
Al-Musharaf, et al. (2021) [13]	Cohort	232 (99 GDM and 133 controls)	Visit 1: weeks 8–12 Visit 2: weeks 24–28	↔ ↔	Visit 1: −Systolic blood pressure, total cholesterol, triglycerides, HDL-cholesterol Visit 2: +HOMA-β It is not specified whether the correlations are considering only the GDM group or the study sample in general.
Saucedo, et al. (2020) [14]	Case-control	65 GDM and 65 controls	At the end of pregnancy	↔	Not reported
Atarod, et al. (2020) [15]	Case-control	37 GDM and 37 controls matched for maternal age and BMI	Weeks 24–28	↓	In GDM: −Impaired blood glucose
Wang, et al. (2020) [16]	Case-control	71 GDM and 66 controls	Weeks 23–28	↓	In GDM: −Fasting plasma glucose, fasting insulin and HOMA-IR.
Francis, et al. (2020) [17]	Cohort	321 (107 GDM and 214 controls)	Visit 1: weeks 10–14 Visit 2: weeks 15–26 Visit 3: weeks 23–31 Visit 4: weeks 33–39	↓ ↓ ↓ ↔	Concentrations at visits 1 and 2: −GDM risk Concentrations at visit 1: −HOMA-IR, glucose, insulin, C-peptide at visit 1 −Triglycerides at visit 2. +HDL-cholesterol at visit 2
Yuan, et al. (2018) [18]	Cohort	359 (86 GDM and 273 controls)	Weeks 16–18	↓	Serum adiponectin levels were a significant predictor of GDM.
Mierzyński, et al. (2018) [19]	Case-control	105 GDM and 55 controls	Weeks 24–28	↓	In all participants: +Omentin −BMI prior to pregnancy, at the time of blood sampling and at the end of pregnancy.
Sweeting, et al. (2018) [20]	Cohort	980 (248 GDM and 732 controls)	Weeks 11–13	↓	Statistical difference was maintained regardless of ethnicity.
Thagaard, et al. (2017) [21]	Cohort	2590 (107 GDM and 2483 without GDM) classified as normal weight (18.5 ≤ BMI ≤ 24.9), moderately obese (30 ≤ BMI ≤ 34.9) and severely obese (BMI ≥ 35).	Weeks 6–14	↓	Adiponectin concentrations were lower in GDM in all BMI groups, although the difference was more pronounced in women with BMI < 35.
Guelfi, et al. (2017) [22]	Cohort	123 (52 GDM and 71 controls)	Visit 1: week 14 Visit 2: week 28 (at the time of diagnosis)	↓ ↓	All participants had a history of GDM in previous pregnancy.
Abell, et al. (2017) [23]	Cohort	103 (25 GDM and 78 controls)	Weeks 12–15	↓	−Fasting glucose, GDM risk. High molecular weight adiponectin was measured.
Leptin					
Tagoma, et al. (2022) [12]	Cohort	213 (60 GDM and 153 controls)	Weeks 23–28	↔	In GDM women, overweight and obesity were associated with higher levels of leptin.
Al-Musharaf, et al. (2021) [13]	Cohort	232 (99 GDM and 133 controls)	Visit 1: weeks 8–12 Visit 2: weeks 24–28	↔ ↔	Visit 1: +Systolic blood pressure, midarm circumference, insulin −Body fat percentage
Saucedo, et al. (2020) [14]	Case-control	65 GDM and 65 controls	At the end of pregnancy	↔	+Pre-pregnancy BMI considering both groups.
Francis, et al. (2020) [17]	Cohort	321 (107 GDM and 214 controls)	Visit 1: weeks 10–14 Visit 2: weeks 15–26 Visit 3: weeks 23–31 Visit 4: weeks 33–39	↑ ↑ ↑ ↔	Concentrations at visits 1 and 2: +GDM risk Concentrations at visit 1: +HOMA-IR, insulin, C-peptide, hsCRP at visit 2.
Sweeting, et al. (2018) [20]	Cohort	980 (248 GDM and 732 controls)	Weeks 11–13	↓	Concentrations are influenced by ethnicity.
Thagaard, et al. (2017) [21]	Cohort	2590 (107 GDM and 2483 without GDM) classified as normal weight (18.5 ≤ BMI ≤ 24.9), moderately obese (30 ≤ BMI ≤ 34.9) and severely obese (BMI ≥ 35).	Weeks 6–14	↑	Leptin concentrations were higher in GDM, but only in women with BMI ≥ 35.
Guelfi, et al. (2017) [22]	Cohort	123 (52 GDM and 71 controls)	Visit 1: week 14 Visit 2: week 28 (at the time of diagnosis)	↔ ↔	All participants had a history of GDM in previous pregnancy.
Zhang, et al. (2017) [24]	Case-control	50 GDM and 50 controls	≥37 weeks	↑	Not reported
Resistin					
Tagoma, et al. (2022) [12]	Cohort	213 (60 GDM and 153 controls)	Weeks 23–28	↔	In GDM women, premature bith was associated with higher levels of resistin.
Al-Musharaf, et al. (2021) [13]	Cohort	232 (99 GDM and 133 controls)	Visit 1: weeks 8–12 Visit 2: weeks 24–28	↔ ↔	Visit 1: +Insulin −Body fat percentage
Saucedo, et al. (2020) [14]	Case-control	65 GDM and 65 controls	At the end of pregnancy	↔	Not reported
Guelfi, et al. (2017) [22]	Cohort	123 (52 GDM and 71 controls)	Visit 1: week 14 Visit 2: week 28 (at the time of diagnosis)	↔ ↔	All participants had a history of GDM in previous pregnancy.
TNF-α					
Peña-Cano, et al. (2021) [25]	Case-control	116 GDM and 115 controls	At the end of pregnancy	↔	+BMI at the end of pregnancy considering both groups.
Al-Musharaf, et al. (2021) [13]	Cohort	232 (99 GDM and 133 controls)	Visit 1: weeks 8–12 Visit 2: weeks 24–28	↔ ↔	Visit 1: −Maternal age Logistic regression adjusted for age and BMI showed that TNF-α levels significantly predict the development of GDM. Significance is lost when other maternal variables are added to the adjustment.
Yin X, et al. (2020) [26]	Case-control	49 GDM and 39 controls	At the end of pregnancy	↑	Not reported
Saucedo, et al. (2020) [14]	Case-control	65 GDM and 65 controls	At the end of pregnancy	↑	+Newborn weight. With predictive value for large for gestational age newborn. +Pre-pregnancy BMI considering both groups. Difference was not maintained after adjustment for maternal age and weight.
Šimják, et al. (2018) [27]	Cohort	24 (12 GDM and 12 controls)	Visit 1: 28–32 weeks Visit 2: 36–38 weeks Visit 3: 6–12 months after delivery	↑ ↑ ↑	Not reported
IL-6					
Peña-Cano, et al. (2021) [25]	Case-control	116 GDM and 115 controls	At the end of pregnancy	↔	+Gestational weight gain considering both groups.
Al-Musharaf, et al. (2021) [13]	Cohort	232 (99 GDM and 133 controls)	Visit 1: weeks 8–12 Visit 2: weeks 24–28	↔ ↔	−Cholesterol/HDL ratio
Abell, et al. (2017) [23]	Cohort	103 (25 GDM and 78 controls)	Weeks 12–15	↔	+BMI at the time of blood sampling +GDM risk
Šimják, et al. (2018) [27]	Cohort	24 (12 GDM and 12 controls)	Visit 1: 28–32 weeks Visit 2: 36–38 weeks Visit 3: 6–12 months after delivery	↔ ↔ ↔	Not reported
IL-8					
Šimják, et al. (2018) [27]	Cohort	24 (12 GDM and 12 controls)	Visit 1: 28–32 weeks Visit 2: 36–38 weeks Visit 3: 6–12 months after delivery	↔ ↔ ↔	Not reported
IL-10					
Šimják, et al. (2018) [27]	Cohort	24 (12 GDM and 12 controls)	Visit 1: 28–32 weeks Visit 2: 36–38 weeks Visit 3: 6–12 months after delivery	↑ ↔ ↔	Not reported
IFN-γ					
Šimják, et al. (2018) [27]	Cohort	24 (12 GDM and 12 controls)	Visit 1: 28–32 weeks Visit 2: 36–38 weeks Visit 3: 6–12 months after delivery	↔ ↔ ↔	Not reported
Adipsin					
Saucedo, et al. (2020) [14]	Case-control	65 GDM and 65 controls	At the end of pregnancy	↑	+Pre-pregnancy BMI considering both groups. The difference was not maintained after adjustment for maternal age and weight.
NGAL					
Saucedo, et al. (2020) [14]	Case-control	65 GDM and 65 controls	At the end of pregnancy	↔	Not reported
Yin, et al. (2020) [26]	Case-control	49 GDM and 39 controls	At the end of pregnancy	↑	In all participants: + Fasting plasma glucose in the second and third trimester, fasting insulin, HOMA-IR, triglycerides and neonatal weight.
Sweeting, et al. (2018) [20]	Cohort	980 (248 GDM and 732 controls)	Weeks 11–13	↑	Multivariate regression analysis showed its value as a significant predictor of GDM. Concentrations are influenced by ethnicity.
NGF					
Saucedo, et al. (2020) [14]	Case-control	65 GDM and 65 controls	At the end of pregnancy	↔	Not reported
MCP-1					
Saucedo, et al. (2020) [14]	Case-control	65 GDM and 65 controls	At the end of pregnancy	↑	+Newborn weight. +Pre-pregnancy BMI considering both groups. Difference was not maintained after adjustment for maternal age and weight.
Abell, et al. (2017) [23]	Cohort	103 (25 GDM and 78 controls)	Weeks 12–15	↔	Not reported
Omentin-1					
Peña-Cano, et al. (2021) [25]	Case-control	116 GDM and 115 controls	At the end of pregnancy	↓	−Pre-pregnancy BMI, BMI at the end of pregnancy and HOMA-IR considering both groups. +HDL considering both groups. The difference was not maintained after adjustment for maternal age, gestational age and BMI.
Francis, et al. (2020) [17]	Cohort	321 (107 GDM and 214 controls)	Visit 1: weeks 10–14 Visit 2: weeks 15–26 Visit 3: weeks 23–31 Visit 4: weeks 33–39	↔ ↔ ↔ ↔	Concentrations at visit 1: −HOMA-IR, insulin, C-peptide and triglycerides at visit 2.
Mierzyński, et al. (2018) [19]	Case-control	105 GDM and 55 controls	Weeks 24–28	↓	In all participants: +Adiponectin. −BMI prior to pregnancy, at the time of blood sampling and at the end of pregnancy. −Risk of preterm birth
Abell, et al. (2017) [23]	Cohort	103 (25 GDM and 78 controls)	Weeks 12–15	↓	−GDM risk
FABP4					
Francis, et al. (2020) [17]	Cohort	321 (107 GDM and 214 controls)	Visit 1: weeks 10–14 Visit 2: weeks 15–26 Visit 3: weeks 23–31 Visit 4: weeks 33–39	↑ ↑ ↑ ↑	Concentrations at visits 1 and 2: +GDM risk Concentrations at visit 1: +Glucose, insulin, C-peptide and hsCRP at visit 2. −HDL-cholesterol at visit 2
Guelfi, et al. (2017) [22]	Cohort	123 (52 GDM and 71 controls)	Visit 1: week 14 Visit 2: week 28 (at the time of diagnosis)	↔ ↔	All participants had a history of GDM in previous pregnancy.
Chemerin					
Francis, et al. (2020) [17]	Cohort	321 (107 GDM and 214 controls)	Visit 1: weeks 10–14 Visit 2: weeks 15–26 Visit 3: weeks 23–31 Visit 4: weeks 33–39	↑ ↑ ↑ ↑	Concentrations at visits 1 and 2: +GDM risk. Concentrations at visit 1: +hsCRP, triglycerides at visit 2.
Guelfi, et al. (2017) [22]	Cohort	123 (52 GDM and 71 controls)	Visit 1: week 14 Visit 2: week 28 (at the time of diagnosis)	↔ ↔	All participants had a history of GDM in previous pregnancy.
Yang, et al. (2017) [28]	Cohort	163 (19 GDM and 144 controls)	Visit 1: weeks 8–12 Visit 2: at around week 31	↓ ↑	A positive association between the risk of GDM and first trimester chemerin levels is reported; however, it does not make sense, as the association should be negative.
sOB-R					
Francis, et al. (2020) [17]	Cohort	321 (107 GDM and 214 controls)	Visit 1: weeks 10–14 Visit 2: weeks 15–26 Visit 3: weeks 23–31 Visit 4: weeks 33–39	↓ ↓ ↓ ↓	Concentrations at visits 1 and 2: −GDM risk. Concentrations at visit 1: −HOMA-IR, glucose, insulin, C-peptide and hsCRP at visit 2. +HDLD at visit 2.
Vaspin					
Francis, et al. (2020) [17]	Cohort	321 (107 GDM and 214 controls)	Visit 1: weeks 10–14 Visit 2: weeks 15–26 Visit 3: weeks 23–31 Visit 4: weeks 33–39	↔ ↔ ↔ ↓	No relevant correlations.
Mierzyński, et al. (2019) [29]	Case-control	153 GDM and 84 controls	Weeks 24–28	↓	In all participants: −Nesfatin-1 levels, BMI prior to pregnancy, BMI at sampling, fasting blood glucose, 1 h and 2 h post-OGTT glucose levels.
RBP-4					
Francis, et al. (2020) [17]	Cohort	321 (107 GDM and 214 controls)	Visit 1: weeks 10–14 Visit 2: weeks 15–26 Visit 3: weeks 23–31 Visit 4: weeks 33–39	Data not shown	Concentrations at visit 1: +LDL-cholesterol and triglycerides at visit 2. The authors did not report RBP-4 concentrations for each visit.
Visfatin					
Abell, et al. (2017) [23]	Cohort	103 (25 GDM and 78 controls)	Weeks 12–15	↔	Not reported
PAI-1					
Tagoma, et al. (2022) [12]	Cohort	213 (60 GDM and 153 controls)	Weeks 23–28	↔	No relevant correlations.
ANGPTL8					
Seyhanli, et al. (2021) [30]	Case-control	45 GDM and 45 controls	Weeks 18–39	↑	+Fasting plasma glucose, fasting plasma insulin, 1 h and 2 h post-load plasma glucose, HOMA-IR, and triglycerides.
Abdeltawab, et al. (2021) [31]	Case-control	109 GDM and 103 controls	Weeks 24–28	↑	+Fasting blood glucose, glycated hemoglobin, LDL-cholesterol, total cholesterol, triglycerides, 1 h and 2 h postprandial blood glucose levels. Multivariate logistic regression showed its value as a significant predictor of GDM.
Gülcü Bulmuş, et al. (2020) [32]	Case-control	30 GDM and 30 controls	Weeks 24–28	↑	+Insulin, C-peptide, and HOMA-IR.
Huang, et al. (2018) [33]	Cohort	474 (88 GDM and 386 controls)	Weeks 12–16	↑	Using multivariable logistic regression, ANGPTL8 levels were related to risk of GDM.
Nesfatin-1					
Çaltekin, and Caniklioğlu (2021) [34]	Case-control	44 GDM and 40 controls	Weeks 24–28	↓	No significant correlations.
Mierzyński, et al. (2019) [29]	Case-control	153 GDM and 84 controls	Weeks 24–28	↓	In all participants: +BMI prior to pregnancy, BMI at sampling, fasting blood glucose, 1 h and 2 h post-OGTT glucose levels.
Ademoglu, et al. (2017) [35]	Case-control	40 GDM and 30 controls	Weeks 24–28	↓	+Gestational age
Zhang, et al. (2017) [24]	Case-control	50 GDM and 50 controls	≥37 weeks	↑	+BMI prior to pregnancy, BMI before delivery, fasting insulin, HOMA-IR and triglycerides. Serum nesfatin-1 was the only independent riskfactor for GDM after adjusting for the BMI before delivery and fasting insulin.
DLK1					
Çaltekin, and Caniklioğlu (2021) [34]	Case-control	44 GDM and 40 controls	Weeks 24–28	↓	+Fasting insulin and HOMA-IR.
Fetuin A					
Jin, et al. (2020) [36]	Nested case-control	135 GDM and 135 controls	Visit 1: 7–13 weeks Visit 2: 25–28 weeks	↑ ↑	The change in fertuin A levels from the first to the second trimester was first found to be associated with the changes in insulin sensitivity and β-cell function, and associated with an increased risk of the development of GDM.
Šimják, et al. (2018) [27]	Cohort	24 (12 GDM and 12 controls)	Visit 1: 28–32 weeks Visit 2: 36–38 weeks Visit 3: 6–12 months after delivery	↔ ↔ ↔	+Uric acid, and CRP. −Creatinine, total bilirubin.
Fetuin B					
Šimják, et al. (2018) [27]	Cohort	24 (12 GDM and 12 controls)	Visit 1: 28–32 weeks Visit 2: 36–38 weeks Visit 3: 6–12 months after delivery	↔ ↔ ↔	+Triglycerides, and CRP. −Total bilirubin.
AFABP					
Zhang, et al. (2017) [24]	Case-control	50 GDM and 50 controls	≥37 weeks	↑	Not reported

GDM, gestational diabetes mellitus; TNF-α, tumor necrosis factor alpha; IL-6, interleukin 6; IL-8, interleukin 8; IL-10, interleukin 10; IFN-γ, interferon-gamma; NGAL, neutrophil gelatinase associated lipocalin; NGF, nerve growth factor; MCP-1, monocyte chemoattractant protein-1; FABP4, fatty acid binding protein-4; sOB-R, soluble leptin receptor; RBP-4, retinol binding protein-4; PAI-1, plasminogen activator inhibitor-1; ANGPTL8, angiopoietin-like protein 8; DLK1, delta like-1; AFABP, adipocyte fatty acid binding protein; HDL-cholesterol, high-density lipoprotein cholesterol; HOMA-β, homeostasis model assessment of β-cell function; HOMA-IR, homeostatic Model Assessment of Insulin Resistance; BMI, body mass index; hsCRP, high-sensitivity C-reactive protein; CRP, C-reactive protein; LDL-cholesterol, low-density lipoprotein cholesterol; OGTT, oral glucose tolerance test. ↑, increased concentrations of the indicated adipokine in GDM compared to controls; ↓, decreased concentrations of the indicated adipokine in GDM compared to controls; ↔, similar concentrations of the indicated adipokine in GDM compared to controls; +, positive and independent correlation between the indicated adipokine and the specified parameter; −, negative and independent correlation between the indicated adipokine and the specified parameter.

**Table 2 ijms-23-06279-t002:** Characteristics and findings of studies comparing adipokine gene or protein expression in adipose tissue between GDM patients and controls at delivery.

Authors (Year)	*n*	Sample and Measurement Type	Adipokine Expression in GDM	Secondary Findings and Comments
Adiponectin				
Dong, et al. (2018) [79]	10 GDM and 27 controls (3 subgroups: 10 obese, 8 overweight and 9 normal weight)	VAT for protein levels	↓	Adiponectin expression was lower in women with GDM than in all subgroups of controls.
Ott, et al. (2018) [80]	25 GDM and 30 controls	VAT for mRNA levels	↓	+Maternal circulating levels of adiponectin −Glucose concentrations at the time of OGTT and at the end of pregnancy.
Telejko, et al. (2010) [81]	20 GDM and 16 controls	VAT for mRNA levels	↓	Not reported
Telejko, et al. (2010) [81]	20 GDM and 16 controls	SAT for mRNA levels	↔	Not reported
Ott, et al. (2018) [80]	25 GDM and 30 controls	SAT for mRNA levels	↓	+Maternal circulating levels of adiponectin
Leptin				
Lappas, et al. (2014) [82]	18 GDM and 28 controls	VAT for mRNA levels	↔	Within the controls, a higher expression of leptin was observed in obese compared to non-obese women
Dong, et al. (2018) [79]	10 GDM and 27 controls (3 subgroups: 10 obese, 8 overweight and 9 normal weight)	VAT for protein levels	↑	No relevant secondary finding
Omentin-1				
Peña-Cano, et al. (2022) [25]	50 GDM and 50 controls	VAT for mRNA levels	↔	No relevant secondary finding
Barker, et al. (2012) [83]	22 GDM and 22 controls	VAT for mRNA and protein levels	↔	In controls, a negative effect of obesity on mRNA and protein levels of omentin-1 was observed. mRNA and protein levels of omentin-1 were measured.
Resistin				
Dong, et al. (2018) [79]	10 GDM and 27 controls (3 subgroups: 10 obese, 8 overweight and 9 normal weight women)	VAT for protein levels	↑	No relevant secondary finding
IL-1β				
Peña-Cano, et al. (2022) [25]	50 GDM and 50 controls	VAT for mRNA levels	↓	No relevant secondary finding
Lappas, et al. (2014) [82]	18 GDM and 28 controls	VAT for mRNA levels	↔	Within the controls, higher expression of IL1-β was observed in obese compared to non-obese women
IL-6				
Peña-Cano, et al. (2022) [25]	50 GDM and 50 controls	VAT for mRNA levels	↓	No relevant secondary finding
IL-1RA				
Peña-Cano, et al. (2022) [25]	50 GDM and 50 controls	VAT for mRNA levels	↓	No relevant secondary finding
IL-10				
Peña-Cano, et al. (2022) [25]	50 GDM and 50 controls	VAT for mRNA levels	↓	After controlling the analysis for gestational age, pre-gestational BMI, and BMI at the end of pregnancy, only IL-10 expression remained significantly lower in women with GDM
TNF-α				
Peña-Cano, et al. (2022) [25]	50 GDM and 50 controls	VAT for mRNA levels	↔	No relevant secondary finding
Rancourt, et al. (2020) [84]	19 GDM and 22 controls	VAT for mRNA levels	↑	+Maternal circulating levels of TNF-α.
Rancourt, et al. (2020) [84]	19 GDM and 22 controls	SAT for mRNA levels	↔	No relevant secondary finding
Dong, et al. (2018) [79]	10 GDM and 27 controls (3 subgroups: 10 obese, 8 overweight and 9 normal weight women)	VAT for mRNA and protein levels	↑	
Lappas, et al. (2014) [82]	18 GDM and 28 controls	VAT for mRNA levels	↔	No relevant secondary finding
SOCS3				
Rancourt, et al. (2020) [84]	19 GDM and 22 controls	VAT for mRNA levels	↑	No relevant secondary finding
Rancourt, et al. (2020) [84]	19 GDM and 22 controls	SAT for mRNA levels	↔	No relevant secondary finding
Visfatin				
Ma, et al. (2010) [85]	20 GDM and 22 controls	SAT for mRNA and protein levels	↔	No relevant secondary finding
Ma, et al. (2010) [85]	20 GDM and 22 controls	VAT for mRNA and protein levels	↔	No relevant secondary finding
Apelin				
Telejko, et al. (2010) [81]	20 GDM and 16 controls	SAT for mRNA levels	↔	No relevant secondary finding
Telejko, et al. (2010) [81]	20 GDM and 16 controls	VAT for mRNA levels	↔	No relevant secondary finding
ADM				
Dong, et al. (2018) [79]	10 GDM and 27 controls (3 subgroups: 10 obese, 8 overweight and 9 normal weight women)	VAT for mRNA levels	↑	The gene expression in VAT of ADM receptor components (CRLR, RAMP2, and RAMP3) was higher in GDM than controls.
Nesfatin-1				
Zhang, et al. (2017) [24]	50 GDM and 50 controls	SAT for protein levels	↑	No relevant secondary finding

GDM, gestational diabetes mellitus; VAT, visceral adipose tissue; SAT, subcutaneous adipose tissue; mRNA, messenger ribonucleic acid; IL-1β, interleukin 1 beta; IL-6, interleukin 6; IL-1RA, interleukin-1 receptor antagonist; IL-10, interleukin 10; TNF-α, tumor necrosis factor alpha; SOCS3, suppressor of cytokine signaling 3; ADM, adrenomedullin; CRLR, calcitonin receptor-like receptor; RAMP, receptor activity-modifying protein; OGTT, oral glucose tolerance test; BMI, body mass index. ↑, increased mRNA and/or protein levels of the indicated adipokine in GDM compared to controls; ↓, decreased mRNA and/or protein levels of the indicated adipokine in GDM compared to controls; ↔, similar mRNA and/or protein levels of the indicated adipokine in GDM compared to controls; +, positive and independent correlation between the indicated adipokine and the specified parameter; −, negative and independent correlation between the indicated adipokine and the specified parameter.

**Table 3 ijms-23-06279-t003:** Characteristics and findings of studies comparing adipokine mRNA or protein levels in placenta and cord blood between GDM patients and controls at delivery.

Authors (Year)	*n*	Sample and Measurement Type	Adipokine Level in GDM	Secondary Findings and Comments
TNF-α				
Yin, et al. (2020) [26]	49 GDM and 39 controls	Placental tissue for mRNA and protein levels	↑	In GDM, increased expression of TNF-α was observed in the placenta compared to the umbilical cord tissue, which was not observed in the control group.
Yin, et al. (2020) [26]	49 GDM and 39 controls	Umbilical cord tissue for mRNA and protein levels	↑	No relevant secondary finding
Yin, et al. (2020) [26]	49 GDM and 39 controls	Cord blood for circulating levels	↓	No relevant secondary finding
Leptin				
Shang, et al. (2018) [100]	105 GDM and 103 controls	Placental tissue for protein levels	↑	A mathematical model including placental levels of leptin, adiponectin and resistin was shown to correlate positively with maternal HOMA-IR.
Manoharan, et al. (2019) [101]	40 GDM and 40 controls	Cord blood for circulating levels	↑	+Ponderal index, Newborn leptin/adiponectin ratio correlated positively with newborn HOMA-IR levels and negatively with HOMA-S
Ortega-Senovilla, et al. (2011) [102]	98 GDM and 86 controls	Cord blood for circulating levels	↔	No relevant secondary finding
Tan, et al. (2021) [103]	95 GDM and 470 controls	Cord blood for circulating levels	↑	Not reported
Shang, et al. (2018) [100]	105 GDM and 103 controls	Cord blood for circulating levels	↑	A mathematical model including umbilical cord concentrations of leptin, adiponectin and resistin was shown to correlate positively with newborn birthweight.
Apelin				
Telejko, et al. (2010) [81]	20 GDM and 16 controls	Placental tissue for mRNA levels	↔	No relevant secondary finding
Aslan, et al. (2011) [104]	30 GDM and 30 controls	Cord blood for circulating levels	↔	No relevant secondary finding
NGAL				
Yin, et al. (2020) [26]	49 GDM and 39 controls	Placental tissue for mRNA and protein levels	↑	+Maternal NGAL concentrations in all of the subjects included in the study. In GDM, increased expression of NGAL was observed in the placenta compared to the umbilical cord tissue, which was not observed in the control group.
Yin, et al. (2020) [26]	49 GDM and 39 controls	Umbilical cord tissue for mRNA and protein levels	↑	No relevant secondary finding
Yin, et al. (2020) [26]	49 GDM and 39 controls	Cord blood for circulating levels	↑	+Maternal NGAL concentrations in all of the subjects included in the study.
Adiponectin				
Shang, et al. (2018) [100]	105 GDM and 103 controls	Placental tissue for protein levels	↓	A mathematical model including placental levels of leptin, adiponectin and resistin was shown to correlate positively with maternal HOMA-IR.
Telejko, et al. (2010) [81]	20 GDM and 16 controls	Placental tissue for mRNA levels	No detectable	Not reported
Manoharan, et al. (2019) [101]	40 GDM and 40 controls	Cord blood for circulating levels	↓	Newborn leptin/adiponectin ratio correlated positively with newborn HOMA-IR levels and negatively with HOMA-S
Shang, et al. (2018) [100]	105 GDM and 103 controls	Cord blood for circulating levels	↓	A mathematical model including umbilical cord concentrations of leptin, adiponectin and resistin was shown to correlate positively with newborn birthweight.
Ortega-Senovilla, et al. (2011) [102]	98 GDM and 86 controls	Cord blood for circulating levels	↓	No relevant secondary finding
Tan, et al. (2021) [103]	95 GDM and 470 controls	Cord blood for circulating levels	↓	Not reported
Resistin				
Yu, et al. (2020) [105]	15 GDM and 15 controls	Placental blood for circulating levels	↑	Not reported
Shang, et al. (2018) [100]	105 GDM and 103 controls	Placental tissue for protein levels	↑	A mathematical model including placental levels of leptin, adiponectin and resistin was shown to correlate positively with maternal HOMA-IR.
Manoharan, et al. (2019) [101]	40 GDM and 40 controls	Cord blood for circulating levels	↔	Not reported
Shang, et al. (2018) [100]	105 GDM and 103 controls	Cord blood for circulating levels	↑	A mathematical model including umbilical cord concentrations of leptin, adiponectin and resistin was shown to correlate positively with newborn birthweight.
Visfatin				
Ma, et al. (2010) [85]	20 GDM and 22 controls	Placental tissue for mRNA and protein levels	↑	+Maternal serum visfatin levels
Manoharan, et al. (2019) [101]	40 GDM and 40 controls	Cord blood for circulating levels	↑	−Total cholesterol, LDL-cholesterol and triglycerides in GDM newborns.
Omentin-1				
Barker, et al. (2012) [83]	22 GDM and 22 controls	Placental tissue for mRNA and protein levels	↔	In controls, a negative effect of obesity on mRNA and protein levels of omentin-1 in placental tissue was observed.
Fetuin A				
Šimják, et al. (2018) [106]	12 GDM and 12 controls	Placental tissue for mRNA levels	↔	No relevant secondary finding
Šimják, et al. (2018) [106]	12 GDM and 12 controls	Cord blood for circulating levels	↓	No relevant secondary finding
RBP-4				
Ortega-Senovilla, et al. (2011) [102]	98 GDM and 86 controls	Cord blood for circulating levels	↑	No relevant secondary finding
Nesfatin-1				
Zhang, et al. (2017) [24]	50 GDM and 50 controls	Cord blood for circulating levels	↑	No relevant secondary finding
Aslan, et al. (2011) [104]	30 GDM and 30 controls	Cord blood for circulating levels	↔	No relevant secondary finding
AFABP				
Ortega-Senovilla, et al. (2011) [102]	98 GDM and 86 controls	Cord blood for circulating levels	↓	+Prepregnancy BMI and maternal leptin levels in all of the subjects included in the study
Vaspin				
Huo, et al. (2015) [107]	30 GDM and 27 controls	Placental tissue for mRNA and protein levels	↔	−Neonatal birth weight
Irisin				
Yuksel, et al. (2014) [108]	20 GDM and 20 controls	Cord blood for circulating levels	↔	No relevant secondary finding

GDM, gestational diabetes mellitus; mRNA, messenger ribonucleic acid; TNF-α; NGAL, neutrophil gelatinase associated lipocalin; RBP-4, retinol binding protein 4; AFABP, adipocyte fatty acid binding protein. HOMA-IR, homeostatic model assessment of insulin resistance; HOMA-S, homeostatic model assessment of insulin sensitivity; LDL-cholesterol, low-density lipoprotein cholesterol; ↑, increased mRNA and/or protein levels of the indicated adipokine in GDM compared to controls; ↓, decreased mRNA and/or protein levels of the indicated adipokine in GDM compared to controls; ↔, similar mRNA and/or protein levels of the indicated adipokine in GDM compared to controls; +, positive and independent correlation between the indicated adipokine and the specified parameter; −, negative and independent correlation between the indicated adipokine and the specified parameter.

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
