# Peer review of "New Insights into Adipokines in Gestational Diabetes Mellitus"

_ijms, 2022, doi:10.3390/ijms23116279_

Round 1
Reviewer 1 Report
The review article will provides many information about the adipokines in gestational diabetes mellitus to readers. On the other hand, it may be difficult to understand because this is only a list of previous publications. Figure 1 is good because it supports readers' comprehensive understanding. How about having some modifications about configuration? For example, further comprehensive figures for circulating adipokines, adipokines in adipose tissues, and adipokines in placenta and umbilical cord are recommended instead of Table 1 and method chapter. In addition, repetition of phrasing like "XX, et al., ..." should be avoided.
Reviewer 2 Report
Comments,
Although the ambitious and interesting title, the present review does not add new information about the role of adipokines in GDM. Most of the references are old and do not cover the last few years during which many papers have been written on GDM and new associated adipokines [e.g., Ruszała M, Novel Biomolecules in the Pathogenesis of Gestational Diabetes Mellitus 2.0. Int J Mol Sci. 2022 Apr 14;23(8):4364; Tagoma A, Plasma cytokines during pregnancy provide insight into the risk of diabetes in the gestational diabetes risk group. J Diabetes Investig. 2022 May 7. doi: 10.1111/jdi.13828; Mallardo M, GDM-complicated pregnancies: focus on adipokines. Mol Biol Rep. 2021 Dec;48(12):8171-8180 ; Tan K, Determinants of cord blood adipokines and association with neonatal abdominal adipose tissue distribution. Int J Obes (Lond). 2022 Mar;46(3):637-645]. Moreover the aim “to compare maternal and umbilical circulating cytokines profiles and the levels of expression of adipokines in VAT, SAT, and placenta in GDM” has not been very successful. Reading the manuscript is difficult to follow since data are presented in a confusing manner. The authors have organized table 1 containing data related maternal circulating adipokines, additional tables reporting data from VAT,SAT, placenta and cord blood would be helpful.
In the present review the authors should make a greater effort and try to give a general picture of the expression of adipokines in the various tissue districts throughout pregnancy/GDM: and how they work, how they interact, and their role in maintaining body homeostasis.
To progress into the knowledge of a given topic (e.g., adipokines in GDM) a review should respond to several questions:
-what we know about the expression in plasma, VAT, SAT, placenta, cord blood in NGT and GDM?
- what we know about their function in the different body districts? Do they exert similar functions? It is possible to provide/hypothesize a network among the different body districts? And how do they impact pregnancy?
Overall, a literature review should clearly summarize the latest data on a given topic, uncover areas in which more research is needed, and also draw conclusions and suggestions for future studies.
